# Improving Local Identifiability in Probabilistic Box Embeddings

**Shib Sankar Dasgupta**[*]
Department of Computer Science
University of Massachusetts, Amherst
ssdasgupta@cs.umass.edu

**Michael Boratko**[*]
Department of Computer Science
University of Massachusetts, Amherst
mboratko@cs.umass.edu

**Dongxu Zhang**
Department of Computer Science
University of Massachusetts, Amherst
dongxu@cs.umass.edu

**Luke Vilnis**
Department of Computer Science
University of Massachusetts, Amherst
luke@cs.umass.edu

**Xiang Lorraine Li**
Department of Computer Science
University of Massachusetts, Amherst
xiang@cs.umass.edu

**Andrew McCallum**
Department of Computer Science
University of Massachusetts, Amherst
mccallum@cs.umass.edu

## Abstract

Geometric embeddings have recently received attention for their natural ability to represent transitive asymmetric relations via containment. Box embeddings, where objects are represented by $n$-dimensional hyperrectangles, are a particularly promising example of such an embedding as they are closed under intersection and their volume can be calculated easily, allowing them to naturally represent calibrated probability distributions. The benefits of geometric embeddings also introduce a problem of local identifiability, however, where whole neighborhoods of parameters result in equivalent loss which impedes learning. Prior work addressed some of these issues by using an approximation to Gaussian convolution over the box parameters, however this intersection operation also increases the sparsity of the gradient. In this work we model the box parameters with min and max Gumbel distributions, which were chosen such that the space is still closed under the operation of intersection. The calculation of the expected intersection volume involves all parameters, and we demonstrate experimentally that this drastically improves the ability of such models to learn.

## 1  Introduction

Geometric embedding models have recently been explored for their ability to learn hierarchies, transitive relations, and partial order structures. Rather than representing objects with vectors, geometric representation models associate domain elements, such as knowledge base queries, images, sentences, concepts, or graph nodes, with objects whose geometry is more suited to expressing relationships in the domain.

Geometric embedding models have used Gaussian densities [25, 1], convex cones, as in order embeddings and entailment cones [24, 10, 5], and axis-aligned hyperrectangles, as in box embeddings

---

[*]Equal Contributions.

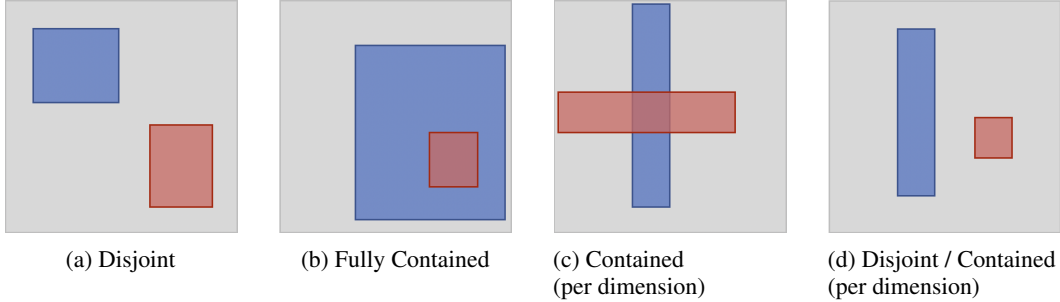

|               |                   |                              |                                      |
|:-------------:|:-----------------:|:----------------------------:|:------------------------------------:|
| (a) Disjoint  | (b) Fully Contained | (c) Contained (per dimension) | (d) Disjoint / Contained (per dimension) |

Figure 1: Settings of parameters which lack local identifiability. For example, any local perturbation of a preserves zero joint probability. For b and c independent local translations of the boxes (for example) preserve the joint probability. Prior work [13] improves only the first case, and even then only to a degree, as it still suffers a lack of local identifiability for settings such as d.

[26, 22, 13] and query2box [21]. Recent work also explores extensions to non-Euclidean spaces such as *Poincaré* embeddings [18] and *hyperbolic entailment cones* [5].

These representations can provide a much more natural basis for transitive relational data, where e.g. entailment can be represented as inclusion among cones or boxes. Additionally, these methods allow for intrinsic notions of an objects breadth of scope or marginal probability, as well as the ability to accurately represent inherently multimodal or ambiguous concepts.

In this work, we focus on the probabilistic box embedding model, which represents the probabilities of binary random variables in terms of volumes of axis-aligned hyperrectangles. While this model flexibly allows for the expression of positively and negatively correlated random variables with complex latent dependency structures, it can be difficult to train due to a lack of local identifiability in the parameter space. That is, large neighborhoods of parameters can give equivalent probabilities to the same events, which leads to gradient sparsity during training (see figure 1). Previous work partially addresses some settings of parameters which lack local identifiability that arise when training box embeddings using Gaussian convolutions [13], but many still remain, leading to a variety of pathological parameter settings, impeding learning.

We generalize the probabilistic box embedding model to a random process model over parametric *families* of box embeddings, which we term the *Gumbel-box processes*. Our approximation to the marginal likelihood of this model has an intuitively pleasing closed form. The resulting model replaces the sparse gradients of the base model's volume calculations with effectively log-smoothed approximations having support on the entire parameter space.

We apply our model to a variety of synthetic problems and demonstrate that the new objective function correctly solves cases that are not solvable by any previous approaches to fitting box models. In real-world data, we demonstrate improved performance on a WordNet completion task and a MovieLens density estimation task.[2]

## 2   Related Work

In addition to the related work discussed in the previous section, most directly relevant is previous work on box embeddings [26, 22, 13], which represent variables as a lattice of axis-aligned hyperrectangles, used to solve problems in density estimation, textual entailment, and graph completion. Very recently, the query2box model [21] has been used to solve problems in knowledge graph reasoning by representing queries as boxes containing sets of answers in a latent space.

This work focuses on improving learning and inference for box models. This was explored in Li et al. [13] by convolving parts of the energy landscape with a Gaussian kernel, similar to approaches in mollified optimization used in machine learning models such as diffusion-trained neural networks

A python package for box embeddings can be found at `https://www.iesl.cs.umass.edu/box-embeddings/`

[15] and mollifying networks [6]. Rather than taking an optimization-based approach, our work introduces a latent random process model which effectively ensembles over a family of box models, similar to various nonparametric random process models such as Gaussian [20] and Dirichlet [16] processes, and Bayesian neural networks [17, 11]. Unlike those models, we rely on latent variables to provide improved credit attribution during training and solve identifiability issues, similar to exploration in reinforcement learning and bandit algorithms [3].

## 3 Background

### 3.1 Probabilistic Box Embeddings

Probabilistic box embeddings, introduced in Vilnis et al. [26], are a learnable compact representation of a joint probability distribution. Variables are associated with elements of the set of $n$-dimensional axis-aligned hyperrectangles,

$$\{x : x = [x_1^\wedge, x_1^\vee] \times \cdots \times [x_d^\wedge, x_d^\vee] \subseteq \mathbb{R}^d\}, \tag{1}$$

called *box embeddings*. Given these box embeddings, a joint probability distribution over these binary variables is defined by treating each box as the indicator of an event set.

**Definition 1** (Indicator random variables). Given a probability space $(\Omega, \mathcal{E}, P)$ and an event $E \in \mathcal{E}$, an *indicator random variable* of $E$ is a random variable $\mathbb{1}_E : \Omega \to \{0, 1\}$ that takes on the value 1 on the set $E$ and 0 otherwise, i.e. $\int_\Omega \mathbb{1}_E dP(\omega) = P(E)$.

Representing each variable with such an indicator enables the model to learn complex interactions between them, because they can overlap in the latent space.

**Definition 2** (Box probability model). Let $(\Omega_{\text{Box}}, \mathcal{E}, P_{\text{Box}})$ be a probability space where $\Omega_{\text{Box}} \subseteq \mathbb{R}^d$. Let $\{\text{Box}(X_i)\}_{i=1}^N$ be a set of boxes in $\Omega_{\text{Box}}$. Each is parameterized by a pair of vectors

$$x^{i,\wedge}, x^{i,\vee} \in \Omega_{\text{Box}} \subseteq \mathbb{R}^d \quad \text{and} \quad \forall j \in \{1, \ldots, d\}, \quad x_j^{i,\wedge} < x_j^{i,\vee}.$$

In terms of these parameters, the boxes are defined as

$$\text{Box}(X_i) = \prod_{j=1}^d [x_j^{i,\wedge}, x_j^{i,\vee}],$$

so that $x^{i,\wedge}$ is a vector of coordinatewise minima for box $i$ and $x^{i,\vee}$ are the coordinatewise maxima. Further, define a set of indicator random variables $\{X_i\}$ for these box-shaped events,

$$X_i : \Omega_{\text{Box}} \to \{0, 1\} = \mathbb{1}_{\text{Box}(X_i)}$$
$$X_i^{-1}(\{1\}) = \text{Box}(X_i).$$

We call the set of random variables $\{X_1, ..., X_N\}$, along with the associated probability space $(\Omega_{\text{Box}}, \mathcal{E}, P_{\text{Box}})$, a *box probability model*.

If each $X_i = f(a)$ for some mapping $f : S \to (\Omega_{\text{Box}} \to \{0, 1\})$ from a finite set $S$ to random variables in the box probability model, we call this a *probabilistic box embedding* of $S$.

**Remark 1.** *The above definition implies that joint probabilities over multiple variables can be calculated by inclusion-exclusion with set intersection and complement over* $\text{Box}(X_i)$*, e.g.*

$$P(X_1 = 1, X_2 = 1, X_3 = 0) = P_{\text{Box}}(\text{Box}(X_1) \cap \text{Box}(X_2) \cap \text{Box}(X_3)^{\mathsf{c}}).$$

The authors observe that, with the addition of the empty set, this set is closed under intersection, a property our model also preserves. In addition, calculating the volume of such elements is trivial, as the operation tensors over dimensions. This enables box embeddings to be learned via gradient descent, and the authors demonstrate the efficacy of these box embeddings to represent graphs, conditional probability distributions, and model entailment.

## 3.2 Local Identifiability

As mentioned previously, in order for the box space to be closed under intersections it is necessary to include the empty set. This seemingly minor point actually belies a deeper issue with learning, namely that the model exhibits a lack of *local identifiability*.

**Definition 3** (Local Identifiability). A set of parameters $\Omega$ is **locally identifiable** if, for all $\theta \in \Omega$, there exists $N(\theta)$, a neighborhood of $\theta$, such that for all $\theta' \in N(\theta)$, $L(x|\theta') \neq L(x|\theta)$.

Probabilistic box embeddings are not locally identifiable, as many settings of their parameters result in equivalent probability distributions. This property was observed in Vilnis et al. [26], where the authors pointed out that large degrees of freedom in the parameter space exhibit this "slackness". For certain forms of structured representation this lack of local identifiability can be seen as a benefit, however the authors also acknowledge that it quickly leads to difficulties with learning methods which rely on local information in the parameter space.

## 3.3 Existing Mitigations

Current work addresses the specific case of pairwise marginals which are incorrectly represented as disjoint. Vilnis et al. [26] introduce a surrogate function which serves to minimize the volume of the smallest containing box. This approach was replaced with a more principled approach in Li et al. [13], where the authors smoothed the hard indicator functions of boxes using Gaussian convolutions and approximated the resulting integral using a softplus function.

Unfortunately, neither of these methods address the many other cases of nonidentifiability. For example, when boxes contain one another (see figure 1b) any small enough translation of the boxes (independently) as well as any scaling of the containing box preserves the volume of the intersection. Full containment is not required, intersections such as those depicted in figure 1c suffer a lack of local identifiability also - consider scaling the red box horizontally and the blue one vertically, and/or translating the boxes independently. Existing methods do not address these situations at all, and furthermore do not even completely address the case of local identifiability for disjoint boxes. For each, boxes which are aligned as depicted in figure 1d will result in equal joint probability throughout any small (indepdendent) vertical translation. Addressing these situations which occur frequently during learning is the main goal of this work.

# 4 Method

In order to mitigate learning difficulties arising from model unidentifiability and the attendant large flat regions of parameter space, we propose to generalize the box probability model to a random process model which ensembles over an entire family of box models.

## 4.1 Gumbel-box process

Computing pairwise or higher-order marginals in the box model involves computing box intersections, which reduces the number of parameters which have access to the gradient signal. An ensemble of box models maintains uncertainty over this intersection, keeping both parameters involved in the computation. Allowing all parameters to contribute to the likelihood of the data in an appropriate way mitigates the aforementioned problems with learning and credit attribution.

A natural way to generalize to a family of box models is to model the parameters of each box as drawn from a Gaussian distribution, which we term the *Gaussian-box process*. This follows Definition 2 in most of the technical details. Since this does not allow us to solve for the expected volume of a box in closed form, we propose an approach using Gumbel random variables, and relegate the details of the Gaussian-box process to Appendix A.

Assume that $\{X_j^{i,\wedge}\}$ and $\{X_j^{i,\vee}\}$ are random variables taking on values in some probability space $(\Omega_{\text{Box}}, \mathcal{E}, P_{\text{Box}})$, representing the per-dimension min and max coordinates of a set of boxes, which are indicator random variables $\{X^i\}$. Under the uniform base measure on $\Omega_{\text{Box}}$, the probability of a

set of variables $T$ taking on the value $1$ is

$$P(T=1) = \prod_{i=1}^{d} \max \big( \min_{X^t \in T} X_j^{t,\vee} - \max_{X^t \in T} X_j^{t,\wedge}, 0 \big), \tag{2}$$

and negated variables can be computed similarly using inclusion-exclusion.

In order to calculate the expected value of (2), it would be desirable to choose a distribution which is $\min$ and $\max$ stable. Such distributions are known as Generalized Extreme Value distributions, an example of which is the Gumbel Distribution

$$f(x; \mu, \beta) = \frac{1}{\beta} \exp(-\tfrac{x-\mu}{\beta} - e^{-\frac{x-\mu}{\beta}}). \tag{3}$$

This distribution is $\max$ stable, i.e. the $\max$ of two such variables also follow a similar distribution, and thus we refer to this as MaxGumbel. By negating the arguments, we therefore obtain a min-stable distribution, which we refer to as MinGumbel. Formally, we have the following:

**Lemma 1.** *If* $X \sim \text{MaxGumbel}(\mu_x, \beta)$, $Y \sim \text{MaxGumbel}(\mu_y, \beta)$, *then*

$$\max(X, Y) \sim \text{MaxGumbel}(\beta \ln(e^{\frac{\mu_x}{\beta}} + e^{\frac{\mu_y}{\beta}}), \beta),$$

*and if* $X \sim \text{MinGumbel}(\mu_x, \beta)$, $Y \sim \text{MinGumbel}(\mu_y, \beta)$, *then*

$$\min(X, Y) \sim \text{MinGumbel}(-\beta \ln(e^{-\frac{\mu_x}{\beta}} + e^{-\frac{\mu_y}{\beta}}), \beta).$$

This motivates the definition of a *Gumbel-box process*. For a probability space $(\Omega_{\text{Box}}, \mathcal{E}, P_{\text{Box}})$, with $\Omega_{\text{Box}} \subseteq \mathbb{R}^d$, the Gumbel-box process is generated as

$$\beta \in \mathbb{R}_+, \quad \mu^{i,\vee} \in \Omega_{\text{Box}} \quad \mu^{i,\wedge} \in \Omega_{\text{Box}}$$
$$X_j^{i,\wedge} \sim \text{MaxGumbel}(\mu_j^{i,\wedge}, \beta), \quad X_j^{i,\vee} \sim \text{MinGumbel}(\mu_j^{i,\vee}, \beta)$$
$$\text{Box}(X^i) = \prod_{j=1}^{d} \left[ X_j^{i,\wedge}, X_j^{i,\vee} \right], \quad X^i = \mathbb{1}_{\text{Box}(X^i)}$$
$$x^1, \dots, x^n \sim P(X^1, \dots, X^n)$$

The definition of the random variables $X^i$ implies that for a uniform base measure on $\mathbb{R}^d$, each $X^i$ is distributed according to

$$X^i \sim \text{Bernoulli}(\prod_{j=1}^{d} (X_j^{i,\vee} - X_j^{i,\wedge})),$$

and the joint distribution over multiple $X^i$ is given by

$$P(T=1, F=0) = P_{\text{Box}}\big( \big( \bigcap_{X^t \in T} \text{Box}(X^t) \big) \cap \big( \bigcap_{X^f \in F} \text{Box}(X^f)^{\text{c}} \big) \big),$$

where $T$ is the subset of variables $\{X^i\}$ taking on the value $1$, $F$ is the subset taking on the value $0$, and $P_{\text{Box}}$ is the base measure on $\Omega_{\text{Box}} \subset \mathbb{R}^d$ used to integrate the volumes of boxes, which we take to be the standard uniform measure in this work. Note that $S^{\text{c}}$ denotes the complement of the set $S$.

**Remark 2.** *Note that the uniform measure can only be a valid probability measure* $P_{\text{Box}}$ *if the base sample space* $\Omega_{\text{Box}}$ *is bounded, and* $X^{i,\wedge}$ *and* $X^{i,\vee}$ *are appropriately constrained to remain within* $\Omega_{\text{Box}}$. *This means that the Gumbel distributions in the Gumbel-box process must be either truncated or censored to* $\Omega_{\text{Box}}$ *for formal correctness. In this work we approximate the expected volume calculation with unconstrained integrals in order to keep a closed form, and elide by abuse of notation. We examine the effect of these approximations in Appendix C.*

The likelihood of an observed data case $(x_1, \ldots, x_n)$ is computed by integrating out the variables $X^{i,\wedge}$ and $X^{i,\vee}$ in equation 2,

$$\log P(x^1, \ldots, x^n | \{\mu^{i,\vee}\}, \{\mu^{i,\wedge}\}, \beta) =$$
$$\log \mathbb{E}[P(x^1, \ldots, x^n, X^{1,\wedge}, X^{1,\vee}, \ldots, X^{n,\wedge}, X^{n,\vee} | \{\mu^{i,\vee}\}, \{\mu^{i,\wedge}\}, \beta)].$$

In this instance, discarding the constraint that $\Omega_{\text{Box}}$ be bounded, calculating the expected value of (2) is tractable under the uniform measure, and we find the following.

**Proposition 1.** *The expected length of an interval $[X, Y]$ where $X \sim \text{MaxGumbel}(\mu_x, \beta)$ and $Y \sim \text{MinGumbel}(\mu_y, \beta)$ is given by*

$$\mathbb{E}[\max(Y - X, 0)] = 2\beta K_0 \left( 2e^{-\frac{\mu_y - \mu_x}{2\beta}} \right). \tag{4}$$

*where $K_0$ is the modified Bessel function of second kind, order $0$.*

The proof of this statement is included in the Appendix B.

Given this formula, we can calculate the expected volume of a box. Since the $X^{i,\wedge}$, $X^{i,\vee}$ are random variables which are $\min$ and $\max$ stable, respectively, the intersection of boxes is also represented by Gumbel random variables, and thus we can calculate the expected volume of intersection.

Let $m(x) = 2\beta K_0 \left( 2e^{-\frac{x}{2\beta}} \right)$. Explicitly, we have

$$\mathbb{E}\left[ \prod_{i=1}^{d} \max\left( \min_{X^t \in T} X_j^{t,\wedge} - \max_{X^t \in T} X_j^{t,\vee}, 0 \right) \right]$$
$$= \prod_{i=1}^{d} m\left( -\beta \underset{X^t \in T}{\text{LogSumExp}}\left( -\frac{\mu_j^{t,\wedge}}{\beta} \right) - \beta \underset{X^t \in T}{\text{LogSumExp}}\left( \frac{\mu_j^{t,\vee}}{\beta} \right) \right)$$

where $T$ is a set of variables taking the value 1, and $\mu^{t,\wedge}$ and $\mu^{t,\vee}$ are the vectors of location parameters for the minimum and maximum coordinates of a box $\text{Box}(X^t)$.

Because $m(x)$ is a smooth upper bound of $\max(0, x)$ (see figure in appendix), and $\text{LogSumExp}$ is a smooth upper bound of $\max$, this formula neatly reflects our intuition over what an ensembled, smooth version of the box model should look like.

We would like to use this formula to compute expected probabilities however we cannot incorporate the constraint that $\Omega_{\text{Box}}$ be bounded and still exactly solve the integral. In practice, we constrain the location parameters of the Gumbel distributions to be within $\Omega_{\text{Box}}$, and most of the probability mass lies inside the region of interest, meaning this gives a reasonable approximation. Further, we use the model to compute conditional probabilities (ratios) instead of absolute probabilities, and thus the unboundedness of the base measure space is less of a problem.

Computing conditional probabilities exactly requires the conditional expectation, $\mathbb{E}[P(x^i, x^j, Z)/P(x^j, Z)]$ which is intractable since it is a quotient. Instead, we approximate it by its first-order Taylor expansion, which takes the form of a ratio of two simple expectations: $\mathbb{E}[P(x^i, x^j, Z)]/\mathbb{E}[P(x^j, Z)]$.

We analyze both of these approximations in Appendix C, and come to two main qualitative conclusions. Firstly, the approximate integral with an unconstrained domain assigns greater expected volume to boxes near the edges of the unit hypercube than occurs when using the exact integrals. It is not clear what effect this has on modeling ability. Secondly, the first-order Taylor approximation of the conditional probability consistently undershoots the true expected ratio, potentially making it easier for the model to assign very small probabilities to events.

## 4.2 Softplus Approximation

We train the embeddings via gradient descent, using KL-divergence loss. The analytic calculation of the expected volume in equation (4) involves a modified Bessel function of the second kind, $K_0$, which is differentiable, with derivative $K_1$. The function itself is essentially exponential as $x$ increases, and the volume function approaches a hinge function as $\beta \to 0$. Our input to this function

is a negative exponential, however, which leads to numerical stability concerns. Inspecting the graph of the exact volume $m$ (see figure in Appendix C), we note the extreme similarity to the softplus function, and find

$$m(x) \approx \beta \log(1 + \exp(x/\beta - 2\gamma)$$

to be a reasonable approximation.

For ease of understanding, we provide a concrete instantiation of our algorithm for learning from pairwise conditional probabilities, incorporating all approximations, in Appendix D.

## 5 Results and Experiments

In this section we compare the proposed model (GumbelBox) to existing baselines, including Order Embeddings [24], Hyperbolic Entailment Cones [5], and the previous state-of-the-art model on these tasks, SmoothBox [13]. In addition, we also compare against an alternative random process model which we term GaussianBox. For GaussianBox, the objective is to minimize an upper bound cost function $\mathbb{E}[f(Z)]$ where the $Z \sim \mathcal{N}(\mu, \Sigma)$. Here, $\mu$ is the parameter of the box embeddings and $\Sigma$ is the uncertainty of each box (a detailed description of the Gaussian-box process can be found in Appendix A). We use a diagonal covariance matrix, and make use of the reparameterization trick for backpropagation. For the GumbelBox model we use the approximate inference method described in section 4.1. Both models are trained using KL-divergence loss (negative log-likelihood).

### 5.1 Ranking Task on Tree-Structured Data

Box embeddings have the representational capacity to naturally embed graphs data with very few dimensions; tree-structured data can even be embedded in just 1 dimension. Despite having sufficient representational capacity, however, existing models struggle to train in such a low-dimensional setting. In this task, we compare the learning capability of our proposed embedding methods with SmoothBox [13] on three different tree structures: the mammal hierarchy of WordNet, a balanced tree of 40 nodes with branching factor 3, and a random tree of 3000 nodes. We use the same training technique as described in Li et al. [13], more details on the dataset for this task are provided in Appendix F.

For each edge $(p, c)$ from a parent $p$ to a child $c$ in the tree we target the conditional probability, $P(p|c) = 1$. For the box models, this is computed as $P(p|c) = \text{Vol}(\text{Box}(c) \cap \text{Box}(p)) / \text{Vol}(\text{Box}(c))$. Our learning objective is to minimize the cross entropy between the predicted conditional probability and ground truth tree labels, thus child boxes should be contained inside their parents. We evaluate the performance on parent and child prediction given tuples from the tree $\mathcal{T}$ by fixing one or the other and ranking all nodes according to the model's predicted conditional probability. We report the mean reciprocal rank (MRR) in Table 1. Further details are in Appendix F.[3]

Table 1: Mean Reciprocal Rank (higher is better, with maximum value 1) in 1 and 2 dimensions for all proposed and baseline box embedding methods on three different tree graphs.

| Trees | # Nodes | SmoothBox | | GaussianBox | | GumbelBox | |
|---|---|---|---|---|---|---|---|
| | | 1d | 2d | 1d | 2d | 1d | 2d |
| **Mammal Hierarchy** | 1182 | 0.851 | 0.927 | 0.865 | 0.927 | **0.934** | **0.981** |
| **Balanced Tree (small)** | 40 | 0.691 | **1.000** | 0.683 | 0.691 | **0.971** | **1.000** |
| **Random Tree (large)** | 3000 | 0.015 | 0.084 | 0.010 | 0.104 | **0.058** | **0.270** |

From Table 1, we observe that our GumbelBox method outperforms both GaussianBox and SmoothBox [13] by a large margin on all three datasets. We empirically observe that the gap is larger in 1-dimension, where learning is even more difficult. In the case of a small 40 node balanced tree, it is easily represented by both SmoothBox and GumbelBox, with both achieving an MRR of 1.0. However, in one dimension SmoothBox could not exceed MRR of 0.691, whereas, GumbelBox achieves MRR 0.971. This can be attributed to a lack of local idenfiability. For example, in order for boxes to cross one another in 1-dimension they must, at some point, be entirely contained inside one another, which is one of the settings mentioned previously which lacks local identifiability. As demonstrated by the significant increase in MRR, GumbelBox helps to alleviate this to a large extent.

The task for the 3000 node tree is much harder, as we train on only the transitive reduction and try to predict edges in the transitive closure. The challenges of this problem were imposed to expose the training difficulties encountered by SmoothBox. The GumbelBox embedding achieves 4 times better performance in 1-d and more than 3 times better performance in the 2-d setting. This empirically validates our claim of improving the local identifiability issue to a large extent.

## 5.2 Example of Local Identifiability Problems

In this section we present an example with 2-dimensional box embeddings where the embedding parameters are initialized in a way which lacks local identifiability. In Figure 2, two boxes are initialized such that they form a cross, and the training objective is to minimize their joint probability. Note that this is problematic whenever *any* subset of dimensions exhibits containment, and would therefore be very likely to occur when using higher dimensions (as is common). Note that any neighborhood of this initialization contains translations of the boxes relative to one another which represent the same joint probability. The SmoothBox model tries to minimize this intersection by making both the boxes skinny. However, GumbelBox is able to avoid this local minimum. We also observe that GaussianBox behaves similarly to SmoothBox. This is because the GaussianBox depends on sampling to overcome the identifiability issue, and the most probable samples are in a neighborhood of the existing parameters. GumbelBox handles this easily, since the analytic calculation of the expected intersection provides the model a more global perspective. We also observe similiar improvments on training when one box contains another in Figure 3.

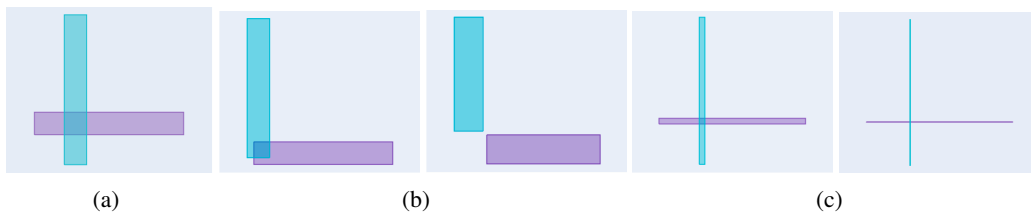

| (a) | (b) | (c) |

Figure 2: (a) Two boxes are initialized as a cross. (b) Second and third pictures demonstrate how GumbelBox is able to train with the objective of minimizing the intersection between the two boxes. (c) Forth and fifth pictures demonstrate how SmoothBox fails to train with the same learning objective.

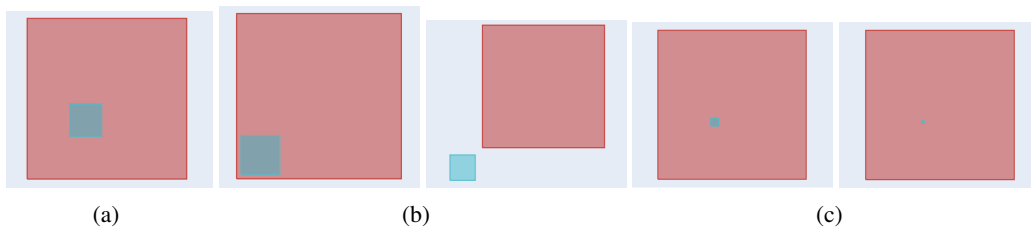

| (a) | (b) | (c) |

Figure 3: (a) Two boxes are initialized such that one box contains another. The objective here is to make them disjoint. (b) Second and third pictures show how GumbelBox is able to train in a desired way w.r.t. the objective. (c) Forth and fifth pictures show how SmoothBox fails to train with the same learning objective.

## 5.3 WordNet

Most of the recent geometric embeddings methods [26, 13, 5, 10, 18] use WordNet [14] to demonstrate the capability of their model to naturally represent real-world hierarchical data. In this section, we demonstrate the performance of our proposed methods on the WordNet noun hierarchy, which consists of $82,114$ entities and $84,363$ edges in its transitive reduction. Following Ganea et al. [5], Patel et al. [19], we train our model on the transitive reduction edges and evaluate on edges in the transitive closure. We report the F1 score on the test set of size $28,838$ edges using $1:10$ negative edges.

Patel et al. [19] improved the performance of box embeddings on this task by penalizing the volume of the embedding when it is greater than a certain threshold. We perform experiments both with

Table 2: Test prediction F1 scores (%) for different methods for WordNet's noun hierarchy.

| Models | Without Regularization | With Regularization |
|---|---|---|
| Order Embedding | 43.0 | - |
| Poincaré Embedding | 28.9 | - |
| Hyperbolic Entailment Cones | 32.2 | - |
| SmoothBox | 45.4 | 60.2 |
| GaussianBox | 43.3 | 58.6 |
| GumbelBox | **51.2** | **62.6** |

and without this regularization. We not only compare our method with SmoothBox [13], but also provide the results from other geometric embedding methods such as Order Embeddings [24], Hyperbolic Entailment Cones [5], Poincaré Embeddings [18]. All models were optimized over equal hyperparameter ranges, and full training details for all models are provided in Appendix G.

We observe from Table 2 that the GumbelBox embedding achieves the best performance in both settings. The boost in performance ($\sim 6$ F1 score) over SmoothBox embeddings is higher when the regularization is not applied. This reinforces our claim that SmoothBox may be encountering settings which lack local identifiability while training, since adding $\ell^2$ regularization on the side-lengths can discourage these plateaus in the loss landscape for SmoothBox. We note that the GaussianBox also performs slightly worse than the SmoothBox in this task as well as in the ranking task (refer Section 5.1).

## 5.4 MovieLens

In this experiment, we predict the preference of a movie A for a user given that he/she has already liked movie B. This dataset was created in the SmoothBox paper [13] by pruning the original MovieLens dataset. We provide the details of the dataset in Appendix H. We compare all the models based on KL divergence, as well as Spearman and Pearson correlation between ground-truth and predicted probabilities. We compare GumbelBox with vanilla matrix factorization and complex bilinear factorization methods [23] as well as geometric embedding methods such as POE [10], Box [26] and SmoothBox [13].

Table 3: Performances of GumbelBox and several baselines on MovieLens.

| | KL | Pearson R | Spearman R |
|---|---|---|---|
| Matrix Factorization | 0.0173 | 0.8549 | 0.8374 |
| Complex Bilinear Factorization | 0.0141 | 0.8771 | 0.8636 |
| POE | 0.0170 | 0.8548 | 0.8511 |
| Box | 0.0147 | 0.8775 | 0.8768 |
| SmoothBox | 0.0138 | 0.8985 | 0.8977 |
| GumbelBox | **0.0120** | **0.9019** | **0.9020** |

We observe from Table 3 that SmoothBox is already achieving a considerable performance boost over the baseline methods. Our model outperforms SmoothBox, although not by a notable margin. We conclude that there is a little scope for improvement over SmoothBox in this task.

## 6 Conclusion and Future work

We presented an approach to improving the parameter identifiability of probabilistic box embeddings, using Gumbel random variables, with an eye towards improved optimization and learning. We demonstrated that our approach fixes several learning pathologies in the naive box model, and advocate it as the default method for training probabilistic box embeddings. Our work uses approximations to the true integrals needed to compute expectations in the random process. Improvements to these approximations present a promising avenue for future work.

## Broader Impact

This work does not present any foreseeable societal consequence.

## Acknowledgments and Disclosure of Funding

We thank our colleagues within IESL, in particular Javier Burroni, for their helpful discussions. We also thank the anonymous reviewers for their constructive feedback. This work was supported in part by the Center for Intelligent Information Retrieval and the Center for Data Science, in part by the Chan Zuckerberg Initiative, in part by the National Science Foundation under Grant No. IIS-1763618, in part by University of Southern California subcontract no. 123875727 under Office of Naval Research prime contract no. N660011924032, and in part by University of Southern California subcontract no. 89341790 under Defense Advanced Research Projects Agency prime contract no. FA8750-17-C-0106. Any opinions, findings and conclusions or recommendations expressed in this material are those of the authors and do not necessarily reflect those of the sponsor.

## Footnotes

[2]Source code and data for the experiments are available at `https://github.com/iesl/gumbel-box-embeddings`.

[3]We use Weights & Biases package [2] to manage our experiments.

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
