[Supplementary Material]

## A  Gaussian-box process

For a probability space $(\Omega_{\text{Box}}, \mathcal{E}, P_{\text{Box}})$, with $\Omega_{\text{Box}} \subseteq \mathbb{R}^d$, the Gaussian-box process is generated as

$$
\begin{aligned}
\mu^i \in \Omega_{\text{Box}}, \quad \sigma^i \in \mathbb{R}_+^d, \quad r \in \mathbb{R}_+^d \\
C^i \sim \mathcal{N}(\mu^i, \sigma^i), \quad X^{i,\vee} = C^i + r^i, \quad X^{i,\wedge} = C^i - r^i, \\
\text{Box}(X^i) = \prod_{j=1}^d \left[ X_j^{i,\wedge}, X_j^{i,\vee} \right], \quad X^i = \mathbb{1}_{\text{Box}(X^i)}, \\
x^1, \ldots, x^n \sim P(X^1, \ldots, X^n)
\end{aligned}
$$

The definition of the random variables $X^i$ implies that for a uniform base measure on $\mathbb{R}^d$, each $X^i$ is distributed according to

$$
X^i \sim \text{Bernoulli}(\prod_{j=1}^d (X_j^{i,\vee} - X_j^{i,\wedge})),
$$

and the joint distribution over multiple $X^i$ is given by

$$
P(T = 1, F = 0) = P_{\text{Box}}\Big( \big( \bigcap_{X^t \in T} \text{Box}(X^t) \big) \cap \big( \bigcap_{X^f \in F} \text{Box}(X^f)^{\mathsf{c}} \big) \Big),
$$

where $T$ is the subset of variables $\{X^i\}$ taking on the value 1, $F$ is the subset taking on the value 0, and $P_{\text{Box}}$ is the base measure on $\Omega_{\text{Box}} \subset \mathbb{R}^d$ used to integrate the volumes of boxes, which we take to be the standard uniform measure in this work. Note that $S^{\mathsf{c}}$ denotes the complement of the set $S$.

**Remark 3.** *Note that the uniform measure can only be a valid probability measure $P_{\text{Box}}$ if the base sample space $\Omega_{\text{Box}}$ is bounded, and $r^i$, $C^i$, $X^i$ are appropriately constrained to remain within $\Omega_{\text{Box}}$. This means that the Gaussian distributions in the Gaussian-box process are actually truncated Gaussians appropriately constrained to $\Omega_{\text{Box}}$, which we elide by abuse of notation.*

To approximately integrate out the latent Gaussian variables, we can backpropagate through sampling using the reparameterization trick [9], which optimizes a lower bound on the log-likelihood of the true model.

## B  Calculation of Expected Volume of a Box

All coordinates will be modeled by independent Gumbel distributions, and thus it is enough to calculate the expected side-length of a box as the expected volume will simply be the product of the expected side-lengths. Let $X \sim \text{MaxGumbel}(\mu_x, \beta)$ be the min coordinate (of a single dimension) and $Y \sim \text{MinGumbel}(\mu_y, \beta)$ be the max coordinate. Let $f_\wedge$ and $F_\wedge$ be the pdf and cdf for MinGumbel, and $f_\vee$ and $F_\vee$ be the pdf and cdf for MaxGumbel. The variance $\beta$ is the same for $X$ and $Y$, and thus we omit the explicit dependence on $\beta$ in the following calculations. The expected

volume is given by

$$E[\max(Y - X, 0)] = \iint_{\{y > x\}} (y - x) f_\wedge(y; \mu_y) f_\vee(x; \mu_x)\, dx dy \tag{5}$$

$$= \int_{-\infty}^{\infty} \int_{-\infty}^{y} y f_\wedge(y; \mu_y) f_\vee(x; \mu_x)\, dx dy -$$
$$\int_{-\infty}^{\infty} \int_{x}^{\infty} x f_\wedge(y; \mu_y) f_\vee(x; \mu_x)\, dy dx \tag{6}$$

$$= \int_{-\infty}^{\infty} y f_\wedge(y; \mu_y) F_\vee(y; \mu_x) dy -$$
$$\int_{-\infty}^{\infty} x f_\vee(x; \mu_x) \left(1 - F_\wedge(x; \mu_y)\right) dx \tag{7}$$

$$= \int_{-\infty}^{\infty} y f_\wedge(y; \mu_y) F_\vee(y; \mu_x) dy -$$
$$\int_{-\infty}^{\infty} x f_\vee(x; \mu_x)\, F_\vee(-x; -\mu_y) dx \tag{8}$$

We note that this is equal to

$$E_Y[Y \cdot F_\vee(Y; \mu_x)] - E_X[X \cdot F_\vee(-X; -\mu_y)], \tag{9}$$

however this calculation can be continued, combining the integrals, as

$$= \int_{-\infty}^{\infty} z f_\wedge(z; \mu_y) F_\vee(z; \mu_x) - z f_\vee(z; \mu_x) F_\vee(-z; -\mu_y)\, dz \tag{10}$$

$$= \int_{-\infty}^{\infty} \frac{z}{\beta}\left(\exp\left(\frac{z - \mu_y}{\beta} - e^{\frac{z - \mu_y}{\beta}} - e^{-\frac{z - \mu_x}{\beta}}\right) - \exp\left(-\frac{z - \mu_x}{\beta} - e^{-\frac{z - \mu_x}{\beta}} - e^{\frac{z - \mu_y}{\beta}}\right)\right) dz \tag{11}$$

$$= \int_{-\infty}^{\infty} \frac{z}{\beta}\left(e^{\frac{z - \mu_y}{\beta}} - e^{-\frac{z - \mu_x}{\beta}}\right) \exp\left(-e^{\frac{z - \mu_y}{\beta}} - e^{-\frac{z - \mu_x}{\beta}}\right) dz \tag{12}$$

We integrate by parts, letting

$$u = z, \qquad dv = \frac{1}{\beta}\left(e^{\frac{z - \mu_y}{\beta}} - e^{-\frac{z - \mu_x}{\beta}}\right) \exp\left(-e^{\frac{z - \mu_y}{\beta}} - e^{-\frac{z - \mu_x}{\beta}}\right) dz, \tag{13}$$

$$du = dz, \qquad v = -\exp\left(-e^{\frac{z - \mu_y}{\beta}} - e^{-\frac{z - \mu_x}{\beta}}\right) \tag{14}$$

where the $dv$ portion made use of the substitution

$$w = e^{\frac{z - \mu_y}{\beta}} + e^{-\frac{z - \mu_x}{\beta}}, \quad dw = \frac{1}{\beta}\left(e^{\frac{z - \mu_y}{\beta}} - e^{-\frac{z - \mu_x}{\beta}}\right) dz. \tag{15}$$

We have, therefore, that (12) becomes

$$= -z \exp\left(-e^{\frac{z - \mu_y}{\beta}} - e^{-\frac{z - \mu_x}{\beta}}\right)\Big|_{-\infty}^{\infty} + \int_{-\infty}^{\infty} \exp\left(-e^{\frac{z - \mu_y}{\beta}} - e^{-\frac{z - \mu_x}{\beta}}\right) dz. \tag{16}$$

The first term goes to 0. Setting $u = \frac{z - (\mu_y + \mu_x)/2}{\beta}$ the second term becomes

$$\beta \int_{-\infty}^{\infty} \exp\left(-e^{\frac{\mu_x - \mu_y}{2\beta}}(e^u + e^{-u})\right) du = 2\beta \int_{0}^{\infty} \exp\left(-2e^{\frac{\mu_x - \mu_y}{2\beta}} \cosh u\right) du. \tag{17}$$

With $z = 2e^{\frac{\mu_x - \mu_y}{2\beta}}$, this is a known integral representation of the modified Bessel function of the second kind of order zero, $K_0(z)$ [4, Eq. 10.32.9], which completes the proof of proposition 1.

## C  Analysis of Approximations

**Domain of integration.** We approximate the box domain $\Omega_{\text{Box}} = [0, 1]^d$ and the densities, which should be restricted to the box domain, by simply restricting the location parameters of the Gumbel

random variables to lie within the box domain and doing an unconstrained integral. That is, we integrate

$$E[\max(Y - X, 0)] = \int_{\mathbb{R}^2} \max(0, y - x) f_\wedge(y; \mu_y) f_\vee(x; \mu_x) \, dxdy.$$

This is inexact because it allows the Gumbel distributions to take on support outside of the domain $[0, 1]$. To properly restrict the Gumbel distributions to $[0, 1]$, we can either form *censored* or *truncated* distributions. The censored distribution puts point masses on the boundary points 0 and 1 corresponding to all of the mass that lies outside of the boundary in the uncensored distribution. This is equivalent to integrating

$$E[\max(Y - X, 0)] = \int_{\mathbb{R}^2} \max(0, \mathrm{clamp}(y) - \mathrm{clamp}(x)) f_\wedge(y; \mu_y) f_\vee(x; \mu_x) \, dxdy,$$

where $\mathrm{clamp}(x) = \min(1, \max(0, x))$. The truncated distribution, on the other hand, multiplies the densities with the indicator function for $[0, 1]$ and renormalizes them to integrate to 1. That is, we integrate

$$E[\max(Y - X, 0)] = \int_{[0,1]^2} \max(0, y - x) \frac{f_\wedge(y; \mu_y)}{z_y} \frac{f_\vee(x; \mu_x)}{z_x} \, dxdy,$$

where $z_x = \int_{[0,1]} f(x; \mu_x) \, dx$.

Figure 4: Expected side length of a Gumbel box with fixed distance between location parameters for min and max, as a function of center position, under the three different integration strategies.

In Figure 4, we show the expected side length under the different integration strategies (computed by numerical quadrature) as a function of center position. We can see that in the unbounded approach that we use in practice, the expected volume of a box does not depend on its center, whereas when properly restricting the Gumbel distributions to $[0, 1]$, the effective volume of boxes towards the edges of the interval becomes smaller. It is not clear what effect this approximation has on model capacity, but it's possible that usage of the exact domains here would allow the model to more easily "pack in" many small probability events.

**Ratio approximation.** When computing conditional probabilities, we approximate the expected ratios under the Gumbel noise distributions by the ratio of expectations. That is, we compute

$$P(X_1 = 1 | X_2 = 1) = \frac{\mathbb{E}[P_{\mathrm{Box}}(\mathrm{Box}(X_1) \cap \mathrm{Box}(X_2))]}{\mathbb{E}[P_{\mathrm{Box}}(\mathrm{Box}(X_2))]},$$

instead of

$$P(X_1 = 1 | X_2 = 1) = \mathbb{E}\Big[\frac{P_{\mathrm{Box}}(\mathrm{Box}(X_1) \cap \mathrm{Box}(X_2))}{P_{\mathrm{Box}}(\mathrm{Box}(X_2))}\Big].$$

The former is a first-order Taylor expansion (linear approximation) of the former. While an expression for the former is not available in closed form and must be computed by Monte Carlo, we can examine

the effect of this approximation by looking at the second order Taylor expansion. The second order Taylor expansion for an expected ratio of two random variables is known to be

$$\mathbb{E}[\frac{A}{B}] \approx \frac{\mathbb{E}[A]}{\mathbb{E}[B]}\Big(1 - \frac{\text{Cov}(A, B)}{\mathbb{E}[A]\mathbb{E}[B]} + \frac{\text{Var}(B)}{\mathbb{E}[B]^2}\Big).$$

In the case where one box's location parameters are contained entirely within the other's with a good separation, we can view the numerator and denominator as independent, and we have

$$\mathbb{E}[\frac{A}{B}] \approx \frac{\mathbb{E}[A]}{\mathbb{E}[B]}\Big(1 + \frac{\text{Var}(B)}{\mathbb{E}[B]^2}\Big),$$

meaning that the first order approximation we use will undershoot the true expected ratio by an amount proportional to the variance of the denominator. Even in the case of correlated numerator and denominator, the quantity in parentheses is always at least equal to one, meaning that the first order approximation consistently undershoots.

The higher the temperature of the boxes, the more the true integral will tend to provide larger conditional probabilities. Monte Carlo experiments support this conclusion. This suggests that the linear approximation might actually help our model by making it easier to assign very small probabilities. However, this may not be true in the situation where the numerator box is not contained within the denominator box by a large margin, and the numerator and denominator become highly correlated.

**Softplus approximation.** We numerically calculate, using the Mathematica software package, the error when approximating

$$2\beta K_0\left(2e^{-\frac{x}{2\beta}}\right) \approx \beta \log\left(1 + \exp(\frac{x}{\beta} - 2\gamma)\right) \tag{18}$$

(where $\gamma$ is the Euler-Mascheroni constant) to be less than $0.0617013\beta$ on for $\frac{x}{\beta} \in [-100, 100]$. Initial explorations with the exact volume suggest that low $\beta$ values perform better, and furthermore suffered from numerical instability (which, given that $K_0$ grows exponentially but our input shrinks exponentially, is unsurprising).

(a) Volume Approximation

(b) Error from Volume Approximation

Figure 5: Graphs depicting error between the exact volume and the softplus approximation

More sophisticated approximations were considered, for example by using a piece-wise combination of the series expansions for $K_0$ at $\infty$ and 0, we can obtain

$$2\beta K_0\left(2e^{-\frac{x}{2\beta}}\right) \approx \begin{cases} 2\beta\sqrt{\frac{\pi}{4}}\exp\left(\frac{x}{4\beta} - 2e^{\frac{-x}{2\beta}}\right)\sum_{k=0}^{n}\frac{\prod_{\ell=0}^{k-1}(2\ell+1)^2}{k!(-16)^k}e^{\frac{xk}{2\beta}} & \text{for } \frac{x}{\beta} \leq \alpha(n), \\ 2\beta\sum_{k=0}^{n}\left(\psi(k+1) + \frac{x}{2\beta}\right)\frac{e^{-\frac{xk}{\beta}}}{(k!)^2} & \text{for } \frac{x}{\beta} > \alpha(n). \end{cases} \tag{19}$$

where $\alpha(n)$ is a transition point chosen to minimize the error.

In practice, however, we found the numerical stability and simplicity of the softplus function to be sufficient for our needs, as demonstrated empirically. We note that, furthermore, this approximation allows for SmoothBox to be viewed as a limiting case of GumbelBox, where the variance for the intersection is taken to zero.

---
**Algorithm 1** Learning from pairwise conditional probabilities
---

**Input:** temperature $\beta$
initialize box parameters $\boldsymbol{\theta} = \{\mu^{\wedge,i}, \mu^{\vee,i}\}$
let $m(x) = \beta \log(1 + \exp(x/\beta - 2\gamma))$, where $\gamma$ is Euler–Mascheroni constant.
**repeat**
    let current parameters $\{\mu^{\wedge,i}, \mu^{\vee,i}\} = \boldsymbol{\theta}_{t-1}$
    receive training probability $P_{\text{train}}(x^i = 1 | x^j = 1)$
    compute $\mu^{ij,\wedge} = -\beta \operatorname{LogSumExp}\left(-\frac{\mu_k^{i,\wedge}}{\beta}, -\frac{\mu_k^{j,\wedge}}{\beta}\right), \mu^{ij,\vee} = \beta \operatorname{LogSumExp}\left(\frac{\mu_k^{i,\vee}}{\beta}, \frac{\mu_k^{j,\vee}}{\beta}\right)$
    compute $P(x^i, x^j = 1) = \prod_{k=1}^d m\left(\mu_k^{ij,\wedge} - \mu_k^{ij,\vee}\right)$
    compute $P(x^j = 1) = \prod_{k=1}^d m\left(\mu_k^{i,\wedge} - \mu_k^{i,\vee}\right)$
    compute $P(x^i = 1 | x^j = 1) = \frac{P(x^i, x^j = 1)}{P(x^j = 1)}$
    compute loss $\ell(\boldsymbol{\theta}_{t-1}) = \operatorname{KL}(P_{\text{train}}(x^i | x^j = 1) | P(x^i | x^j = 1))$
    update $\boldsymbol{\theta}_{t-1}$ with gradient $\nabla \ell(\boldsymbol{\theta}_{t-1})$
**until** CONVERGED$(\boldsymbol{\theta}_t, \boldsymbol{\theta}_{t-1})$

---

# D  Algorithm

For ease of understanding, we provide a concrete instantiation of our algorithm, Algorithm 1, for learning from pairwise conditional probabilities, incorporating all approximations.

# E  Gumbel Box Visualizations

(a) Probability of a point $x$ being contained inside a Gumbel box for different values of $\beta$.

(b) Intersection of two Gumbel boxes, where the inner green distribution represents the probability of a point $x$ being contained inside both.

Figure 6: Visualizations of Gumbel boxes

(a) SoftBox
(temp=1.0)

(b) GumbelBox
(variance=1.0)

(c) Mixed GumbelBox
(variance=0.3, temp=1.0)

Figure 7: Densities of overlap with a square of size $\varepsilon$

(a) SoftBox
(temp=1.0)

(b) GumbelBox
(variance=1.0)

(c) Mixed GumbelBox
(variance=0.3, temp=1.0)

Figure 8: Contour plots of overlap with a square of size $\varepsilon$

(a) SoftBox
(temp=1.0)

(b) GumbelBox
(variance=1.0)

(c) Mixed GumbelBox
(variance=0.3, temp=1.0)

Figure 9: Density plots of overlap with a square of size $\varepsilon$

# F Further details on Experiment 5.1

## F.1 Ranking Task

We describe the ranking task more formally in this section. For a tree $\mathcal{T}$ with set of nodes $\mathcal{N}$, we minimise the following objective function.

$$\min_{\theta} \sum_{(p,c) \in s(\mathcal{TC}) \cup \mathcal{TC}^-} -y \log \mathrm{Prob}(p|c) - (1-y) \log(1 - \mathrm{Prob}(p|c)), \tag{20}$$

Where, $s(\mathcal{TC})$ is a subset of the transitive closure($\mathcal{TC}$) of $\mathcal{T}$ and the negative samples are drawn randomly from the set -

$$\mathcal{TC}^- = \{(p',c)|p' \in \mathcal{N}, (p',c) \notin \mathcal{TC}\} \cup \{(p,c')|c' \in \mathcal{N}, (p,c') \notin \mathcal{TC}\}.$$

The label $y$ takes the value 1 when $(p,c) \in \mathcal{TC}$ and 0 when $(p,c) \in \mathcal{TC}^{-1}$. Here $\theta$ is the set of parameter for the box embeddings.

In case of child prediction, we calculate the prediction score (i.e., the conditional probability $Prob(\cdot|child)$ for all the nodes in $\mathcal{N}$ given the fixed parent from the observed tuple $(parent, child)$. Then we find the rank of that tuple amongst all possible negative tuple of the form $(parent, \cdot)$. Similarly we calculate the rank for parent prediction by keeping the child fixed and ranking the observed parent amongst all possible negative corresponding to the tuple $(\cdot, child)$.

## F.2 Dataset

We use the following three tree-structured data for this experiment.

1. Balanced tree: The tree consists of 40 nodes with branching factor 3 and depth 4. The whole transitive closure of 102 edges were used for training. The ranking task is performed on whole transitive closure.

2. Mammal Hierarchy of WordNet: Number of entities are 1182. The whole transitive closure of 6542 edges were used for training. MRR is calculated on a subset of size 3441 edges which are sampled randomly from the transitive closure.

3. Random tree: This 3000 node tree was generated randomly using networkx. We train only on the transitive reduction of 2999 edges. The MRR is calculated on a 4920 edges which are randomly sampled from the whole transitive closure.

**Hyperparameter range:** For all the models, we use Bayesian hypermeter search with Hyperband algorithm [12]. The hyperparameter ranges are $\beta \in [0.001, 3]$, $lr \in [0.0005, 1]$, batch size $\in \{8096, 2048, 1024, 512, 256\}$, number of negative samples $\in \{2, 5, 10, 20, 25, 40, 70\}$. The best hyperparameters corresponding to each model including the baselines are provided in the *configs* folder of the source code provided.

# G Further details on Experiment 5.3

We use the same binary cross entropy based approach as described in Equation 20 in Section 5.1. However, we use random negative sampling for this task. We keep the dimensions for all the non box-embedding baselines to be 10. Box emebddings are parameterised as *min* and *side length* vectors of 5 dimensions each. For the GaussianBox, the covariance matrix for the Gaussian noise is considered to be diagonal matrix of same dimension as the min vector. The baseline results are as reported in [19].

**Hyperparameter range:** For all the models, we use Bayesian hypermeter search with Hyperband algorithm [12]. The hyperparameter ranges are $\beta \in [0.001, 3]$, $lr \in [0.0005, 1]$, batch size $\in \{1024, 512, 2048, 8096, 16192\}$, number of negative samples(integer) $\in [2, 100]$, regularization weight $\in [0.00001, 0.005]$. The best hyperparameters corresponding to each model including the baselines can be found in the *configs* folder of the source code provided.

# H  Further details on Experiment 5.4

In this section we briefly describe the data curation process by [13] for this task. First, all user-movie pairs with rating higher than 4 was collected and then the dataset is further pruned using the popularity of the movie (it must have more than 100 rating). Here, the model would predict the $P(MovieA|MovieB)$ and the gold labels are generated by dividing the co-occurrence frequency count of movie A and B by the frequency count of movie B. We use the exact same split of 100k/10k/10k provided by [13].

We follow the evaluation setting in [13] for all MovieLens experiments, where models are evaluated every 50 iterations on the validation set, and optimization is stopped if the best development set score fails to improve after 200 evaluations. The best model is then used to score the test set. We use $\beta = 0.01$, softplus temperature = 1.0 with 50 dimensional boxes, batch size 128 and Adam optimizer (learning rate 1e-3) for GumbelBox in our reported results.

# I  Flickr Experiment

We also evaluate GumbelBox on a sentence level entailment task using Flickr, which is an image caption dataset of 45million images. We use the dataset split from [10] in order to perform fair comparison between models. The models are trained to regress to the given ground-truth probabilities calculated using frequency, using KL-divergence loss, following the same model architecture as [13].

We report KL divergence loss and Pearson correlation on the full test data, unseen pairs (caption pairs which are never occur in training data) and unseen words(caption pairs which has at least one word never occur in training data)). As shown in Table 10, we get comparable results compared to SmoothBox and Box.

Table 10: Test KL & pearson correlation score between predicted probabilities and ground truth probabilities for Flickr caption entailment task.

| Dataset / Model | Full test data | | Unseen pairs | | Unseen words | |
|---|---|---|---|---|---|---|
| | KL | Pearson R | KL | Pearson R | KL | Pearson R |
| **POE** | 0.031 | 0.949 | 0.048 | 0.920 | 0.127 | 0.696 |
| **Box** | 0.020 | 0.967 | 0.025 | 0.957 | 0.050 | 0.900 |
| **SmoothBox** | 0.018 | 0.969 | 0.024 | 0.957 | 0.036 | 0.917 |
| **GumbelBox** | 0.020 | 0.969 | 0.024 | 0.960 | 0.035 | 0.910 |

### I.0.1  Experiments details

Model structure: Same as other geometric embedding such as [10, 26, 13], we encode each sentence by a LSTM model [7], then pass the sentence embedding to two feed-forward neural networks to encode minimum and maximum coordinates of the GumbelBox.

Model parameter: We use the same experiment setting as in Li et al. [13] for Flickr. The models are evaluated every epoch for 5 epochs. The best model is saved for the lowest dev KL loss. We then evaluate the best model on test data. The resulting best performing model has the following parameters: $\beta = 1e - 4$, temperature $= 1.0$, learning rate $= 5e - 5$, batch size $= 512$, dropout$= 0.5$, embedding size$= 300$, optimizer=Adam [8]