[Reviews · NeurIPS 2020]

Review 1

Summary and Contributions: The author present a new tool for representing box embedding to solve identifiability issues with current box embedding. By using an ensemble of probabilistic box embedding, this provides a smoother gradient and helps the convergence of the algorithm. They show efficient closed form operation for the gumbel box embeddings. Experiments on a a simple toy dataset clearly exposes the advantage of this new methods compared to existing box embedding approaches. They also expose failure modes of other methods. Experiments on MovieLens shows minor improvements.

Strengths: The authors show a clear improvement over existing methods on box embeddings. The method is clearly explained and closed form solutions are elegant. While limited in scope, this work provide an interesting solution for geometric embeddings and could help this tool grow in popularity.

Weaknesses: Line 179-185: Those 2 approximations are discussed a bit too vaguely. It feel like it is just swept under the rug. Elaborating more on the impact of these approximations and where it could fail. While the resulting method is simple and seems to be stable, the level of mathematic might be out of reach for many reader in our community. Hence, I would encourage the author to provide a clear algorithm section for ease of implementation. Also a simple and well written python package on github available through e.g. pip install would really help to expand the scope. ==== POST DISCUSSION EDIT ===== Well conducted rebuttal. No major flas to be addressed.

Correctness: Gumble box has 3 parameters per dimensions while conventional embeddings have only 1. What is the number of parameters for other approach in Table 2 and 3 ? SmoothBox uses 2 or 3? Are the results directly comparable?

Clarity: Overall the paper is clear, but it could be improved. * A figure similar to Figure 1 could be in the intro to better explain the identifiability issue from the beginning. * Starting with Gaussian-box (4.1) and then showing that there is no closed form was confusing to me. I don't find that 4.1 helps understanding 4.2 it feels more like a waste of cognitive energy to get there and then we have to start over. I see however that it can be useful as a baseline in experiments. So I would suggest putting most of it in appendix and stating "we explore a gaussian version but since there is not closed form, we propose a gumble version" Minors: Line 78: where where Line 99: L is not defined Figure1: Forth <- Fourth

Relation to Prior Work: I am not savvy in the box embedding world, so it is hard to assess if references are missing. The number of references is under the average, but the Related Work section puts well other works in perspective.

Reproducibility: Yes

Additional Feedback:


Review 2

Summary and Contributions: This work proposes a method to mitigate the local identifiability problem of box embeddings, whereby the authors propose modeling the box parameters via min/max Gumbel distributions, which the authors show is more appropriate for joint (intersection) computations.

Strengths: The paper appears well written, succinct, and clearly describes the problem of local identifiability when learning box embeddings. Further, the paper clearly explains the derivations, approximations, and assumptions of their work. The problem statement appears solid, the motivation is apparent, the proposed methodology seems appropriate, and the derivations look to be carefully worked through and reasoned. The empirical evaluations are relatively small, but seeing as how this is a more theoretically driven and motivated problem, I think that their synthetic demonstrations and empirical checks are reasonable.

Weaknesses: As mentioned above, the empirical evaluations are limited: it would be a stronger benefit to the community if this work were evaluated across more problem settings, and ideally, the experimental code were released for easy reproducibility. That being said, I believe the authors provided sufficient evidence that their Gumbel distribution selection sufficiently solves the problems the paper lays out.

Correctness: The claims, methods, and empirical methodology appear correct.

Clarity: The paper is relatively straightforward to follow and well written.

Relation to Prior Work: Yes: in section 3.3 and throughout section 4 the authors clearly describe the previous approaches used to mitigate the local identifiability problems.

Reproducibility: Yes

Additional Feedback: UPDATE: I appreciate the author's response and agree with their updates.


Review 3

Summary and Contributions: Post rebuttal: The authors answered my questions (and pointed out other areas of the paper that explained some of my confusions.) I raised my score by 1 to 7. ------------------------- The goal of this paper is to improve a class of embeddings called box embeddings. The idea in box embeddings is to represent objects as (higher-dimensional) boxes and to use geometric principles, such as intersections, to enforce structure. One of the downsides of this approach is that optimization can be hard, partially due to a lack of identifiability. The authors introduce a generalization where the embeddings are modeled by a random process with distriibutions selected to improve local identifiability. Specifically, Gaussian and Gumbel distribution-based approaches. In this case the edges for each box come from the corresponding distribution. The nice thing about the Guebl approach is the ability to compute in a closed-form way the average volume of the box (here the mean is over the ensemble). The authors then use this principle to train embeddings on some of the typical datasets (wordnet and movielens), showing that there's considerable improvements in performance on a variety of settings.

Strengths: - In general this is an important problem, and the authors identify one of the issues with a popular approach and make a lot of progress on fixing it. - The empirical results seem quite good. I'm not an expert in what's the typical best-in-class performance for various models on these tasks, but nevertheless the comparisons look good. - Good illustrative examples in simple cases (i.e., the problems with lack of identifiability in Fig 1 and Fig 2).

Weaknesses: - It's not clear how much of an improvement this paper is over Li et al, the paper where smoothing box embeddings was first proposed. It would have been good to have a more comprehensive comparison with that paper, at least in terms of the properties of the method. The performance improvements are clear. - The paper doesn't do a great job explaining specifically how the local identifiability issue is improved. There's illustration and occasional discussion of cases, but no rigorous arguments that are made.

Correctness: This is largely an empirical paper that improves a class of techniques. The methodology used for the empirical results is fine.

Clarity: For the most part, the paper is well-written. It also have pretty good illustrations to help the reader understand the arguments.

Relation to Prior Work: For the most part, yes, but there are specific cases where this discussion should be beefed up (as explained above).

Reproducibility: Yes

Additional Feedback: A few more comments and suggestions here: - I would have liked to know more about the wall clock time involved in training these different embeddings. Does the ensemble approach slow anything down? These would be pretty useful details to know. - The overall organization of the paper would be improved if the algorithms were separated from the flow of the paper and put into a box, so that they don't have to be inferred from the details. - As mentioned above, it's not actually that clear (to me) how the current approach ensures that local identifiability is corrected. I can see the correction in Figure 1, but expressing it algebraically would help me understand here. You could write down the sets of parameters that lead to similar loss for these cases and how the ensemble mitigates this.


Review 4

Summary and Contributions: This paper is concerned with the mitigating local identifiability issues in probabilistic box embeddings. Probabilistic box embedding model is used to represent the probabilities of binary variables in terms of volumes of axis-aligned hyperrectangles. These probability distributions can be used to express specific relations between entities such as hierarchies, partial orders, and lattice structures. Learning box embeddings using gradient-based methods are not straightforward due to unidentifiability of such models. Non-identifiability means that the likelihood is not affected for whatever infinitesimal change in parameter space is made, hence relaxation of the boxes is required. In this paper, the definition of identifiability issues is not limited only to unidentifiablity of the model, but also includes issues like an optimization procedure following a spurious gradient direction. E.g. when one box contains another, but they are supposed to be disjoint, the gradient-based optimization procedure will try to decreases the area of the inner box instead of separating the two boxes. Instead of just employing relaxation of the boxes, as it's been done in the previous work the authors proposed an exciting method of using ensembles of the boxes that effectively still provides such relaxation but also improves "credit attribution" during training and thus solves additional local identifiability issues. The effectiveness of the proposed method is shown using artificial and WordNet/MovieLens datasets.

Strengths: I found the proposed method that uses nonparametric-random-process-like ensembling to mitigate the local identifiability issue to be quite exciting. Ensembling nature of the GumbelBox model provides a global perspective on the optimized task and analytically computed expected intersection makes it an efficient method.

Weaknesses: I can't see any critical weaknesses in this work.

Correctness: To the best of my knowledge derivations appear to be correct.

Clarity: The paper is well-written and structured clearly. However, I would still include the conclusion section, to make it easier for the readers who just want to skim through the paper. Also, line 99 I would suggest using mathcal font for L and adding that it means likelihood function. Typo line 239: "expectated" should be "expected".

Relation to Prior Work: The authors clearly discussed how their contribution is related to prior work.

Reproducibility: Yes

Additional Feedback:

[Author Response · NeurIPS 2020]

We thank the anonymous reviewers for their feedback, which we will incorporate into future drafts. We are grateful for the many typographical corrections and suggested edits. As to the main concerns raised by the reviewers, in the interest of brevity we have paraphrased them below in bold, and provided itemized responses.

- **A similar figure to Figure 1 should be in the introduction, to better explain identifiability (Reviewer #1)**

  We agree and will add this in the final version.

- **The algorithm(s) should be broken out into a clear algorithm box for ease of parsing / implementation. (Reviewer #1, #3)**

  We agree and will add this in the final version.

- **It would be useful to provide a Python implementation. (Reviewer #1, #2)**

  For reproducibility, implementation of all models compared in this paper is included with our submission. We are also planning to publish a pip-installable package for box embeddings which will include GumbelBox, SmoothBox, and the original Box Embeddings.

- **Do we use comparable amounts of parameters for vector vs. box-based models? (Reviewer #1, #2)**

  For fair comparison we have used the same number of parameters for all models in experiments 3 and 4, (Table 2 & 3). Note: Gumbel box only has two parameters per entity per dimension, not three - the scale parameter is a global hyperparameter. The experimental code included with the submission contains all hyperparameter settings used.

- **GaussianBox section is confusing to include prominently (Reviewer #1)**

  We thank the reviewer for suggesting these required re-structures to improve clarity. We agree with all of it, and will definitely incorporate them on the final version of the paper.

- **What properties does GumbelBox have compared to SmoothBox from Li et al.? (Reviewer #3)**

  Gumbel box (with the softplus approximation) can actually be interpreted as a generalization of SmoothBox with a nonzero temperature on the intersection operation. This is mentioned in the appendix, but we will make this more clear in the camera ready.

  Adding this nonzero temperature improves learning over SmoothBox in a number of settings (ie. the cases mentioned in the synthetic experiments) by improving local identifiability. Even in scenarios without identifiability issues, GumbelBox has denser gradients, since all parameters of the boxes being intersected contribute to the intersection. This is always beneficial, but particularly so for cases where multilple boxes are intersected, a point we can highlight more in the final version.

- **How does wall-clock time compare to the smoothed box model and other embeddings? (Reviewer #3)**

  Since we compute the expected intersection / volume in closed form rather than by sampling, the overall training time is not appreciably slower. For example, in case of WordNet (Table-2), 150 epochs take around 35 minutes for SmoothBox and 41 minutes for GumbelBox with the same hyperparamters (batch size of 256, embedding dimension of 5, and number of negative samples 16) on GeForce RTX 2080 Ti (averaged over 20 runs).

- **Include a conclusion section (Reviewer #4)**

  We will include a conclusion and future work section in the final version.

- **Approximations on lines 179-185 are discussed too vaguely, please elaborate on possible failure modes. (Reviewer #1)**

  We agree that, even though the empirical results are strong, these approximations could use a bit more discussion. We will add a discussion and visualize graphically the impact of these approximations and edge cases where they could be inaccurate.

- **Make the statements regarding local identifiability clearer and more rigorous. (Reviewer #3)**

  While the local identifiability improvements are discussed in the paper, Figures 4 and 5 in the appendix give more intuition to why the GumbelBox model has improved local identifiability compared to SoftBox. We will move these figures to the main body of the work, which may help clarify the argument.

  The paper also identifies certain specific classes of parameters for which GumbelBox is locally identifiable and SmoothBox is not, which we demonstrate in the initial synthetic experiments, however we agree that more formal statements can be made. For the camera ready we can include a rigorous characterization of these classes and proof of this fact.

  We can also prove that in 1-dimension, or when the per-dimension ratios are fixed, the GumbelBox model is entirely locally identifiable. (Note: all local identifiability is modulo a global translation of the parameters.)

[Meta-Review · NeurIPS 2020]

All reviewers agree that this is a solid contribution, improve the current technique for box embeddings. My recommendation is to accept. Please take the reviewers' comments into account in preparing the final version of the paper.